# Stress-Induced Sleep Dysregulation: The Roles of Astrocytes and Microglia in Neurodegenerative and Psychiatric Disorders

**DOI:** 10.3390/biomedicines13051121

**Published:** 2025-05-06

**Authors:** Ángel R. Rábago-Monzón, Juan F. Osuna-Ramos, David A. Armienta-Rojas, Josué Camberos-Barraza, Alejandro Camacho-Zamora, Javier A. Magaña-Gómez, Alberto K. De la Herrán-Arita

**Affiliations:** 1Facultad de Medicina, Universidad Autónoma de Sinaloa, Culiacán 80019, Mexico; 2Doctorado en Ciencias en Biomedicina Molecular, Facultad de Medicina, Universidad Autónoma de Sinaloa, Culiacán 80019, Mexico; 3Facultad de Ciencias de la Nutrición y Gastronomía, Universidad Autónoma de Sinaloa, Culiacán 80019, Mexico

**Keywords:** astrocytes, microglia, neuroinflammation, sleep homeostasis, neurodegenerative diseases

## Abstract

Stress and sleep share a reciprocal relationship, where chronic stress often leads to sleep disturbances that worsen neurodegenerative and psychiatric conditions. Non-neuronal cells, particularly astrocytes and microglia, play critical roles in the brain’s response to stress and the regulation of sleep. Astrocytes influence sleep architecture by regulating adenosine signaling and glymphatic clearance, both of which can be disrupted by chronic stress, leading to reduced restorative sleep. Microglia, activated under stress conditions, drive neuroinflammatory processes that further impair sleep and exacerbate brain dysfunction. Additionally, the gut–brain axis mediates interactions between stress, sleep, and inflammation, with microbial metabolites influencing neural pathways. Many of these effects converge on the disruption of synaptic processes, such as neurotransmitter balance, synaptic plasticity, and pruning, which in turn contribute to the pathophysiology of neurodegenerative and psychiatric disorders. This review explores how these cellular and systemic mechanisms contribute to stress-induced sleep disturbances and their implications for neurodegenerative and psychiatric disorders, offering insights into potential therapeutic strategies targeting non-neuronal cells and the gut–brain axis.

## 1. Introduction

Sleep is a fundamental physiological process essential for maintaining cognitive, emotional, and physical health. Sleep is characterized as a structured pattern of alternating non-rapid eye movement (NREM) and rapid eye movement (REM) sleep throughout the night, typically cycling every 90 min. NREM sleep consists of three stages (N1, N2, and N3) ranging from light to deep sleep. N1 is the transition from wakefulness to sleeping, marked by slowed brain activity and muscle relaxation. N2 is characterized by sleep spindles and K-complexes, which are thought to protect sleep and support memory consolidation. N3, or slow-wave sleep (SWS), is the deepest stage, dominated by delta waves, and it is crucial for physical restoration and immune function. REM sleep, in contrast, is marked by rapid eye movement, vivid dreaming, and increased brain activity resembling wakefulness, while the body remains paralyzed to prevent acting out dreams. The regulation of this architecture involves complex interactions among brain regions such as the hypothalamus, brainstem, and thalamus, and it is modulated by neurotransmitters including GABA, acetylcholine, norepinephrine, serotonin, and adenosine. This sleep–wake cycle is synchronized with the body’s internal circadian rhythm, which is regulated by the suprachiasmatic nucleus (SCN) of the hypothalamus. Circadian rhythms are entrained by zeitgebers (German for “time-givers”), the most potent of which is light, followed by others such as temperature, food intake, and social cues. Light signals received through the retina are transmitted to the SCN, which acts as the brain’s master clock. The SCN regulates melatonin release from the pineal gland and orchestrates daily physiological rhythms, helping to align sleep architecture with the external environment. At the molecular level, this synchronization is mediated by core circadian clock genes such as CLOCK and BMAL1, which activate the transcription of PER and CRY genes. As PER and CRY proteins accumulate, they inhibit the CLOCK-BMAL1 complex, forming a transcriptional–translational feedback loop that generates a ~24 h cycle. This molecular rhythm governs the timing of sleep stages, influencing when and how NREM and REM sleep occur throughout the night. Disruptions to this system, whether through irregular light exposure or stress-induced changes in gene expression, can lead to misaligned sleep architecture and reduced sleep quality [1].

Stress-induced sleep dysregulation is defined as a disruption in the normal structure, timing, and restorative functions of sleep due to acute or chronic exposure to stress. This condition manifests in various forms, including difficulties in falling asleep (insomnia), frequent awakenings, non-restorative sleep, or changes in the balance of REM and NREM sleep stages [1]. It is particularly significant because it establishes a harmful cycle: stress disrupts sleep, and poor sleep further exacerbates the physiological and psychological impacts of stress [2]. Over time, this can lead to chronic neuroinflammation, cognitive impairments, and heightened vulnerability to neurodegenerative and psychiatric disorders such as Alzheimer’s disease (AD), Parkinson’s disease (PD), major depression, and anxiety disorders [3,4]. 

Stress-induced sleep disturbances are increasingly recognized as critical factors in the pathogenesis of various neurodegenerative and psychiatric disorders [3]. However, the underlying cellular and molecular mechanisms remain incompletely understood, particularly with regard to the contributions of non-neuronal cells such as astrocytes and microglia.

Chronic stress disrupts the delicate balance of astrocytic and microglial activity, leading to neuroinflammation, oxidative stress, and impaired synaptic connectivity. These stress-induced glial dysfunctions are intimately linked to disturbances in sleep homeostasis, creating a bidirectional feedback loop that exacerbates neural dysfunction. Emerging evidence suggests that astrocytes and microglia contribute not only to the onset and maintenance of sleep but also to the pathological consequences of sleep disturbances in neurodegenerative and psychiatric conditions [5,6].

Astrocytes and microglia play indispensable roles in maintaining central nervous system (CNS) homeostasis by regulating neural function, metabolic processes, and neuroimmune responses. Astrocytes, the most abundant glial cells, are highly versatile and contribute to synaptic support, metabolic regulation, and extracellular homeostasis. They are closely associated with synapses, where they regulate synaptic transmission and plasticity by releasing gliotransmitters, buffering extracellular ions, and reabsorbing neurotransmitters such as glutamate. This dynamic interaction enables astrocytes to modulate synaptic activity and maintain optimal neuronal communication. Additionally, astrocytes play a fundamental role in the glymphatic system, which facilitates the clearance of neurotoxic waste, such as amyloid beta and tau proteins, during sleep. Dysfunction of this system, often resulting from chronic stress or poor sleep quality, leads to the accumulation of toxic metabolites, which is a hallmark of neurodegenerative diseases. Moreover, astrocytes regulate neurotransmitter recycling, particularly glutamate, which is critical for maintaining balanced excitatory signaling. Dysregulated astrocytic activity may contribute to excitotoxicity, impairing sleep architecture and exacerbating neurological and psychiatric symptoms [6].

Microglia, the resident immune cells of the CNS, are equally critical for maintaining homeostasis. In their resting state, microglia continuously survey the microenvironment, ensuring neuronal integrity and responding to subtle changes. They play an essential role in synaptic pruning, a process vital for synaptic plasticity and proper neural circuit development. Microglia also participate in immune surveillance, releasing cytokines and chemokines to regulate inflammatory responses and protect against pathogens or injuries. However, under chronic stress or pathological conditions, both astrocytes and microglia can become dysregulated, contributing to neuroinflammation, oxidative stress, and synaptic dysfunction. These disruptions not only compromise CNS homeostasis but also exacerbate the risk of developing neurodegenerative and psychiatric disorders. Chronic activation of microglia, often due to stress or disrupted sleep, can lead to a pro-inflammatory state characterized by the release of cytokines and chemokines. This neuroinflammatory environment is detrimental to sleep quality and is associated with heightened risks of depression, anxiety, and cognitive decline [7]. Furthermore, excessive microglial activation can interfere with synaptic integrity, leading to long-term disruptions in brain function and behavior [8,9].

In this review, we aim to provide a comprehensive analysis of the roles of astrocytes and microglia in stress-induced sleep dysregulation. We will examine their physiological contributions to sleep regulation, explore how chronic stress alters their functions, and discuss the implications of these alterations for neurodegenerative and psychiatric disorders. Additionally, we will highlight cutting-edge research on glial-specific therapeutic interventions and future directions for understanding the interplay between stress, sleep, and glial biology. By elucidating these complex interactions, this review seeks to advance our understanding of the cellular underpinnings of sleep disorders and identify novel targets for therapeutic intervention.

Astrocytes and microglia are increasingly recognized as active participants in sleep regulation, not merely as support cells but as dynamic regulators of neural activity, inflammation, and waste clearance. Their dysfunction can directly impact sleep homeostasis and contribute to the progression of diseases such as AD, PD, depression, and anxiety disorders [10].

Recognizing the convoluted roles of astrocytes and microglia in sleep disorders provides valuable insights into the bidirectional relationship between sleep dysregulation and disease pathology. This understanding paves the way for novel therapeutic strategies that target glial cells to restore sleep homeostasis and mitigate disease progression. For instance, interventions aimed at modulating astrocytic activity or dampening microglial-driven neuroinflammation hold promise for improving sleep quality and addressing the underlying mechanisms of neurodegenerative and psychiatric conditions. Thus, elucidating the contributions of these glial cells is not only important for understanding sleep biology but also for developing targeted therapies for a wide range of brain disorders.

## 2. Stress and Sleep Dysregulation: Mechanistic Insights

The hypothalamic–pituitary–adrenal (HPA) axis is a neuroendocrine system that orchestrates the body’s response to stress while also playing a crucial role in sleep regulation and circadian rhythmicity. This axis is governed by the hypothalamus, which integrates environmental and physiological signals to maintain homeostasis. The paraventricular nucleus (PVN) of the hypothalamus serves as the central regulator of the HPA axis, receiving input from the limbic system, brainstem, and suprachiasmatic nucleus (SCN), the primary circadian pacemaker [11,12].

Under normal conditions, the PVN releases corticotropin-releasing hormone (CRH) and arginine vasopressin (AVP), which stimulate the anterior pituitary to secrete adrenocorticotropic hormone (ACTH). ACTH then acts on the adrenal cortex to promote the synthesis and release of glucocorticoids (cortisol in humans, corticosterone in rodents). These glucocorticoids follow a circadian rhythm, with peak secretion in the early morning to promote wakefulness and nadir levels during the night to facilitate sleep. The rhythm of cortisol secretion is tightly regulated by the SCN via autonomic and hormonal pathways to ensure synchronization with the sleep–wake cycle [13].

However, chronic stress leads to hyperactivation of the HPA axis, resulting in excessive and prolonged glucocorticoid release [14]. Elevated cortisol disrupts sleep homeostasis by interfering with the neurochemical systems that regulate sleep architecture. CRH, the initial mediator of HPA axis activation, has direct wake-promoting effects by stimulating arousal-related neural circuits in the locus coeruleus (LC) and increasing noradrenaline release. CRH also suppresses slow-wave sleep (SWS) and promotes REM sleep fragmentation. Concurrently, cortisol suppresses the release of melatonin from the pineal gland, further impairing circadian synchronization and reducing sleep efficiency [12].

At the molecular level, chronic glucocorticoid exposure alters gene expression in sleep-regulating brain regions, including the prefrontal cortex, hippocampus, and amygdala. One of the primary targets is the glucocorticoid receptor gene (NR3C1), which mediates cellular responses to cortisol; sustained activation can lead to the downregulation or desensitization of this receptor, disrupting feedback mechanisms in the hypothalamic–pituitary–adrenal (HPA) axis. FKBP5, a glucocorticoid-responsive gene that inhibits receptor activity, is often upregulated under chronic stress, contributing to stress-related sleep disturbances and psychiatric vulnerability. Additionally, glucocorticoids impact the expression of core circadian clock genes such as PER1, PER2, CRY1, and CRY2, disrupting the timing and synchronization of sleep–wake cycles [15]. Glucocorticoid receptor (GR) activation leads to the transcriptional repression of brain-derived neurotrophic factor (BDNF), impairing synaptic plasticity and weakening the sleep-promoting functions of these brain structures. Additionally, prolonged cortisol exposure upregulates pro-inflammatory cytokines such as interleukin-6 (IL-6), tumor necrosis factor-alpha (TNF-α), and interleukin-1β (IL-1β), which disrupt sleep homeostasis by interfering with the balance of γ-aminobutyric acid (GABA) and glutamate neurotransmission. IL-1β and TNF-α, in particular, enhance glutamatergic excitotoxicity and reduce the inhibitory tone required for slow-wave sleep initiation [15].

Chronic HPA axis dysregulation also disrupts the SCN, which is responsible for maintaining circadian rhythms. Normally, the SCN entrains peripheral clocks in various tissues via rhythmic oscillations of core clock genes such as CLOCK, BMAL1, PER, and CRY. However, chronic glucocorticoid excess alters the expression of these genes, leading to circadian misalignment between the central SCN clock and peripheral clocks in metabolic and immune tissues. This desynchronization manifests as sleep fragmentation, reduced REM latency, and increased nocturnal awakenings [16,17].

Beyond its direct effects on neurons, stress-induced HPA axis hyperactivation also impacts glial function, further contributing to sleep disturbances. Astrocytes, which regulate extracellular ion balance and neurotransmitter clearance, become reactive under prolonged glucocorticoid exposure, leading to excessive calcium signaling and the disruption of sleep-dependent metabolic processes such as the glymphatic clearance of amyloid-beta and tau proteins. Microglia, the immune sentinels of the brain, adopt a pro-inflammatory phenotype under chronic stress, increasing the release of cytokines that impair synaptic function and disrupt neural circuits involved in sleep regulation [7,8].

The interaction between the HPA axis, circadian rhythms, and glial dysfunction creates a vicious cycle where stress-induced sleep disturbances further exacerbate neuroinflammation, oxidative stress, and cognitive impairments [18]. These disruptions are strongly implicated in the pathogenesis of neurodegenerative diseases (Alzheimer’s and Parkinson’s) and psychiatric disorders (depression, PTSD, and anxiety), where chronic stress exposure accelerates disease progression by amplifying sleep disturbances and synaptic dysfunction [19,20].

## 3. Neuroinflammation and Sleep Architecture: The Impact of Chronic Stress on Sleep Stages and Neuroinflammatory Signaling

Chronic stress induces a persistent neuroinflammatory state that disrupts sleep architecture by altering glial function, cytokine signaling, and neurotransmitter balance [21]. Under physiological conditions, sleep regulation relies on finely tuned neuroimmune chemistry, where inflammatory mediators such as interleukin-1β (IL-1β), tumor necrosis factor-alpha (TNF-α), and prostaglandins promote sleep by modulating neuronal excitability and synaptic plasticity [22,23]. These cytokines act on the ventrolateral preoptic nucleus (VLPO), a critical sleep-promoting region in the hypothalamus, by enhancing GABAergic inhibition of wake-promoting nuclei, including the LC, tuberomammillary nucleus (TMN), and dorsal raphe nucleus (DRN). However, chronic stress disrupts this balance, leading to sustained microglial and astrocytic activation, excessive pro-inflammatory cytokine release, and maladaptive changes in sleep–wake regulation [24].

One of the most prominent effects of chronic stress-induced neuroinflammation is the disruption of NREM sleep, particularly SWS, which is essential for memory consolidation and neural recovery [25,26]. Elevated levels of IL-1β and TNF-α induce hyperexcitability in cortical and thalamic glutamatergic circuits, impairing the generation of synchronized delta wave oscillations (0.5–4 Hz), which are characteristic of deep sleep [27,28]. Additionally, reactive astrocytes exhibit dysregulated adenosine metabolism, reducing extracellular adenosine availability, thereby weakening sleep pressure and increasing sleep fragmentation [28,29] (Figure 1). This combination of cortical hyperactivity and reduced sleep drive results in shallow and less restorative sleep.

In addition to NREM alterations, chronic neuroinflammation profoundly affects REM sleep, a stage crucial for emotional regulation and synaptic remodeling [30]. Sustained microglial activation suppresses cholinergic transmission in the pedunculopontine and laterodorsal tegmental nuclei (PPT/LDT), regions that are essential for REM sleep generation. Simultaneously, increased CRH release from the HPA axis hyperactivates the amygdala, further delaying REM onset and fragmenting REM episodes [31,32,33]. This impairment in REM sleep contributes to emotional dysregulation, exacerbating anxiety and depressive symptoms commonly observed in chronic stress conditions. Additionally, neuroinflammation-driven oxidative stress damages monoaminergic neurons in the raphe nuclei, disrupting serotonin homeostasis and further destabilizing REM sleep [34] (Figure 1). Oxidative stress refers to an imbalance between the production of reactive oxygen species (ROS) and the body’s ability to neutralize them through antioxidant defense mechanisms. This imbalance leads to cellular damage, the activation of inflammatory pathways, and impaired mitochondrial function.

Another hallmark of stress-induced neuroinflammation is increased sleep fragmentation and insomnia, which stem from excessive excitatory signaling and synaptic dysfunction. Chronically elevated levels of nuclear factor kappa B (NF-κB), a key transcription factor regulating inflammatory responses, enhance glutamatergic activity, reducing the stability of sleep–wake transitions. Upon NF-κB activation by stress signals, pro-inflammatory cytokines (such as IL-1β and TNF-α), or oxidative stress, NF-κB translocates to the nucleus, where it alters the expression of genes involved in neuroinflammation and neural excitability. This heightened inflammatory state disrupts the normal function of sleep-regulating brain regions, such as the hypothalamus and brainstem, leading to fragmented or shallow sleep. Moreover, NF-κB interferes with the expression of core circadian clock genes like BMAL1 and CLOCK, contributing to circadian misalignment and the destabilization of sleep architecture. By increasing arousal-related signaling and impairing the consolidation of NREM and REM sleep, NF-κB activity promotes sleep–wake rhythm instability, which is commonly observed in conditions such as chronic stress, depression, and neurodegenerative disorders. Furthermore, peripheral and central interleukin-6 (IL-6) upregulation disrupt hypothalamic orexinergic circuits, increasing arousal and reducing overall sleep efficiency [35,36,37]. The combination of these molecular changes leads to hyperarousal states, characterized by frequent nocturnal awakenings and prolonged sleep latency (Figure 2).

Chronic stress and glucocorticoid release
○Prolonged stress stimulates the release of glucocorticoids, which dysregulate glial function and impair glymphatic clearance.○This results in the accumulation of neurotoxic proteins (e.g., amyloid-beta and tau aggregates), which are normally cleared during sleep.
Glial dysfunction and neuroinflammation
○Dysfunctional astrocytes and microglia enter a pro-inflammatory state, releasing cytokines that further disrupt sleep architecture, as seen in the hypnogram at the top.○Sleep fragmentation exacerbates glial dysfunction, creating a self-reinforcing loop.Synaptic pruning and neuronal damage
○Chronic neuroinflammation leads to abnormal synaptic pruning, weakening essential neuronal connections.○This results in neuronal damage, reducing cognitive function and increasing vulnerability to neurodegenerative diseases (e.g., Alzheimer’s and Parkinson’s) and psychiatric disorders (e.g., depression, anxiety, and schizophrenia).
A vicious cycle of stress and sleep dysregulation
○Poor sleep quality worsens HPA axis dysregulation, leading to sustained stress and further neuronal dysfunction.○Over time, this cycle promotes cognitive decline, mood disorders, and an increased risk of age-related neurodegeneration.


This figure highlights the critical interaction between stress, sleep, and brain health, emphasizing the need for interventions targeting stress reduction and sleep restoration to prevent long-term neurological consequences.

All figures were made in Adobe Illustrator (Illustrator 2025).

Beyond its immediate effects on sleep, chronic neuroinflammation establishes a self-perpetuating cycle that exacerbates neurodegenerative and psychiatric disorders. Impaired sleep homeostasis prevents the clearance of toxic protein aggregates via the glymphatic system, increasing the accumulation of β-amyloid and tau, which are implicated in AD. Additionally, prolonged microglial activation leads to excessive synaptic pruning and neurotoxicity, further impairing cognitive function (Figure 2). The intersection of chronic stress, neuroinflammation, and sleep dysregulation thus represents a key pathological mechanism underlying AD, PD, depression, and PTSD [38,39].

## 4. Astrocytic Control of Sleep Homeostasis

Astrocytes play a fundamental role in regulating sleep homeostasis through their involvement in glymphatic clearance, synaptic pruning, and neurotransmitter metabolism. These highly specialized glial cells orchestrate sleep-related neural processes by modulating extracellular ion balance, neurotransmitter availability, and metabolic waste clearance, all of which are essential for cognitive function and neuroprotection [6,8,11,38,39]. Their dynamic responses to sleep–wake transitions are mediated by intracellular calcium signaling, gliotransmitter release, and interactions with neuronal circuits, positioning astrocytes as key regulators of both sleep stability and recovery.

One of the most critical astrocytic functions in sleep homeostasis is their role in glymphatic clearance, a cerebrospinal fluid (CSF)-dependent system that removes metabolic waste from the brain. During NREM sleep, astrocytes regulate the expansion of perivascular spaces by controlling aquaporin-4 (AQP4) channel activity on their endfeet, which facilitates the convective flow of interstitial fluid [40,41,42]. Astrocytes regulate AQP4 channels through several mechanisms that are crucial for maintaining brain water homeostasis. Calcium signaling within astrocytes, often triggered by glutamate activation at NMDA receptors, modulates the trafficking and insertion of AQP4 channels into the cell membrane. Additionally, intracellular signaling pathways such as MAPK and PI3K/Akt can enhance AQP4 expression on the cell surface in response to changes in osmotic pressure or neuroinflammatory stimuli. Astrocytes express two isoforms (AQP4-M1 and AQP4-M23) which differ in their distribution within the cell, and their regulation is influenced by local signals like cytokines and growth factors. The cytoskeletal protein GFAP also plays a role in positioning AQP4 channels in astrocytic endfeet, and alterations in GFAP expression, such as during reactive gliosis, can disrupt AQP4 trafficking. These coordinated processes ensure proper water balance in the brain, impacting functions like synaptic activity, waste clearance, and edema regulation. This process enhances the clearance of neurotoxic proteins, such as β-amyloid and tau, which, when accumulated, contribute to neurodegenerative diseases like Alzheimer’s. Chronic sleep deprivation impairs AQP4 polarization at the perivascular endfeet, reducing glymphatic efficiency and promoting protein aggregation, neuroinflammation, and cognitive decline [31,32].

Astrocytes also influence synaptic pruning, a process essential for neural plasticity and cognitive maintenance. Through the release of complement proteins (C1q, C3, and C4) and interactions with microglia, astrocytes tag weak or redundant synapses for elimination, a function that is particularly active during sleep [43,44,45,46]. This activity is driven by sleep-dependent changes in calcium wave propagation and ATP release, which regulate the engulfment of excess synaptic connections. Under chronic stress conditions, astrocytes exhibit aberrant pruning activity, leading to either excessive synaptic loss (as seen in depression and neurodegeneration) or insufficient synaptic refinement (which contributes to hyperexcitability and psychiatric disorders such as schizophrenia) [47,48,49].

In addition to structural and metabolic regulation, astrocytes modulate the sleep–wake cycle through their interactions with the adenosinergic and glutamatergic systems. Adenosine, a key somnogenic neuromodulator, accumulates extracellularly as a byproduct of astrocytic ATP metabolism, progressively increasing sleep pressure during wakefulness [49]. Astrocytes control adenosine levels by regulating their synthesis, degradation, and uptake via equilibrative nucleoside transporters (ENTs). When sleep pressure reaches a critical threshold, adenosine acts on A1 receptors (A1Rs) in the basal forebrain and cortex, inhibiting wake-promoting neurons and facilitating sleep onset [50,51]. However, under conditions of chronic stress, glucocorticoid-induced metabolic dysregulation disrupts astrocytic ATP metabolism, leading to insufficient adenosine accumulation, prolonged wakefulness, and increased sleep fragmentation [52].

Astrocytes further contribute to neurotransmitter metabolism, particularly in the regulation of glutamate homeostasis, which is crucial for maintaining sleep stability. Astrocytes uptake excess glutamate via the excitatory amino acid transporters (EAAT1 and EAAT2), converting it into glutamine, which is then recycled to neurons for GABA synthesis. During sleep, this glutamate–GABA cycling is fine-tuned to promote cortical inhibition and facilitate slow-wave oscillations. However, chronic stress-induced astrocytic dysfunction leads to impaired glutamate clearance, resulting in hyperexcitability, reduced slow-wave activity, and increased susceptibility to insomnia and mood disorders [52,53,54].

## 5. Stress-Induced Dysregulation of Glial Functions

Chronic stress profoundly disrupts the homeostatic functions of astrocytes and microglia, leading to widespread neural dysfunction that exacerbates sleep disturbances, neuroinflammation, and synaptic instability. The persistent activation of the HPA axis under chronic stress results in elevated glucocorticoid levels, which impair the regulatory roles of astrocytes and microglia, shifting them toward a maladaptive state. This stress-induced glial dysregulation alters glutamate homeostasis, oxidative balance, calcium signaling, and immune surveillance, contributing to the pathophysiology of neurodegenerative and psychiatric disorders (Figure 2).

Astrocytes play an essential role in maintaining synaptic integrity through glutamate clearance, redox homeostasis, and calcium signaling. However, under chronic stress, elevated cortisol levels disrupt astrocytic glutamate transporters, particularly EAAT2 (also known as GLT-1) [55], leading to excess extracellular glutamate accumulation. This results in glutamate excitotoxicity, where excessive activation of N-methyl-D-aspartate (NMDA) receptors on neurons causes intracellular calcium overload, mitochondrial dysfunction, and neuronal damage [56]. The failure of astrocytes to efficiently remove glutamate contributes to persistent cortical hyperexcitability, reduced SWS, and increased susceptibility to insomnia and mood disorders [57,58,59].

In parallel, chronic stress induces oxidative stress in astrocytes by upregulating NADPH oxidase (NOX) and downregulating antioxidant defenses such as glutathione (GSH) synthesis [60]. Excessive reactive oxygen species (ROS) production damages astrocytic membranes, disrupts gap junctional communication, and weakens their ability to support neurons [61,62,63]. This oxidative burden further impairs the function of the glymphatic system, reducing the clearance of toxic proteins like β-amyloid and tau, thereby increasing the risk of neurodegenerative diseases [64].

Astrocytic calcium signaling, which is essential for coordinating neuronal–glial interactions and regulating sleep homeostasis, is also dysregulated by chronic stress. Under normal conditions, astrocytic calcium waves influence gliotransmitter release (ATP, D-serine, and glutamate) to modulate synaptic activity [65]. However, stress-induced glucocorticoid receptor (GR) overactivation in astrocytes perturbs IP3 receptor-mediated calcium mobilization, impairing their ability to synchronize neural oscillations necessary for sleep stability [52,66]. This disruption weakens astrocytic control over sleep–wake transitions, leading to fragmented sleep and heightened stress reactivity [67].

On the other hand, microglia, the resident immune cells of the CNS, respond to environmental stressors by shifting between homeostatic and reactive states. While transient microglial activation is beneficial for clearing debris and maintaining synaptic function, chronic stress drives sustained microglial hyperactivation, leading to excessive pro-inflammatory cytokine release and aberrant synaptic pruning [68]. One of the primary consequences of chronic stress-induced microglial activation is the excessive production of pro-inflammatory cytokines, including TNF-α, IL-1β, and IL-6. These cytokines disrupt sleep architecture by interfering with the VLPO, a major sleep-promoting region in the hypothalamus. IL-1β and TNF-α inhibit GABAergic neurons in the VLPO, reducing their inhibitory tone on wake-promoting regions such as the locus coeruleus and orexinergic hypothalamic neurons. This shift prolongs wakefulness, increases sleep fragmentation, and reduces SWS and REM sleep, contributing to cognitive deficits and emotional dysregulation (Figure 1) [69,70]. Furthermore, a study indicates that microglia modulate sleep via P2Y12–G_*i*_ signaling, elevated intracellular Ca^2+^ levels, and reduced norepinephrine. These findings feature its crucial function in the interaction between the brain’s immune system and sleep regulation [47].

In addition to promoting neuroinflammation, chronic stress alters microglial phagocytic activity, leading to aberrant synaptic pruning. Under normal physiological conditions, microglia refine synaptic connections by removing weak or redundant synapses, a process regulated by complement proteins (C1q, C3, and C4). However, stress-induced microglial hyperactivation leads to excessive complement-mediated synaptic engulfment, particularly in the prefrontal cortex and hippocampus, regions critical for cognitive and emotional regulation. This aberrant pruning weakens synaptic networks, reducing cognitive flexibility and increasing vulnerability to disorders such as major depressive disorder (MDD), PTSD, and Alzheimer’s disease (Figure 2). Furthermore, prolonged microglial activation results in a shift from a neuroprotective (M2) to a neurotoxic (M1) phenotype, characterized by the overexpression of inducible nitric oxide synthase (iNOS), ROS, and inflammatory prostaglandins. This neurotoxic state exacerbates neuronal damage, weakens blood–brain barrier integrity, and perpetuates a cycle of neuroinflammation-driven sleep disturbances [71,72,73].

## 6. Psychiatric Disorders and Stress-Induced Sleep Dysregulation

The dysregulation of astrocytic and microglial functions due to chronic stress and sleep disturbances has profound implications for neurodegenerative diseases. Sleep is essential for synaptic plasticity, metabolic homeostasis, and the clearance of toxic proteins, and its disruption accelerates neurodegeneration. Stress-induced glial dysfunction exacerbates glymphatic impairment, neuroinflammation, and oxidative stress, contributing to the pathophysiology of AD, PD, Huntington’s disease (HD), and amyotrophic lateral sclerosis (ALS).

AD is characterized by the accumulation of β-amyloid (Aβ) plaques and hyperphosphorylated tau, processes that are tightly regulated by sleep and glial activity. Astrocytes play a key role in glymphatic clearance, a brain-wide waste disposal system that removes toxic proteins during sleep. As mentioned before, the glymphatic system is most active during SWS, where astrocytic AQP4 channels facilitate the flow of CSF along perivascular spaces, clearing metabolic waste [7]. Chronic stress and sleep disturbances impair AQP4 polarization at the astrocytic endfeet, reducing glymphatic function and leading to the accumulation of Aβ and tau aggregates. Sleep fragmentation further disrupts glymphatic flow, exacerbating cognitive decline and neuronal toxicity [13,33]. Additionally, astrocytic dysfunction under chronic stress results in glutamate excitotoxicity, which damages hippocampal neurons and accelerates memory impairment in AD [74]. Microglia also play a pivotal role in Aβ clearance through phagocytosis. However, chronic stress and prolonged sleep loss lead to microglial priming, a state of heightened reactivity characterized by excessive production of pro-inflammatory cytokines (IL-1β, IL-6, and TNF-α) [75,76]. These cytokines promote neuroinflammation, impair synaptic plasticity, and further inhibit microglial-mediated Aβ clearance, creating a self-perpetuating cycle of sleep dysfunction and neurodegeneration [77,78].

PD is marked by dopaminergic neurodegeneration in the substantia nigra pars compacta (SNpc) and is strongly associated with REM sleep behavior disorder (RBD), a parasomnia where individuals physically act out dreams due to loss of muscle atonia [79,80]. Astrocytes and microglia contribute significantly to the neuroinflammatory environment that drives PD progression and sleep disturbances. Chronic stress-induced astrocytic dysfunction impairs glutamate clearance, leading to excessive activation of subthalamic nucleus (STN) neurons, which disrupts basal ganglia circuitry and exacerbates motor and sleep disturbances in PD [81,82,83]. Additionally, astrocytes regulate dopamine metabolism through interactions with monoamine oxidase (MAO-B), and their dysfunction under stress results in dopaminergic neurotoxicity and worsening of PD symptoms [84,85]. Microglia in PD exhibit a chronic pro-inflammatory phenotype (M1 state), releasing TNF-α, IL-1β, and reactive oxygen species (ROS) that exacerbate neurodegeneration in the SNpc. Research suggests that microglia also play a crucial role in sleep regulation by modulating CX3CR1 (which mediates the migration, adhesion, and retention of leukocytes) expression in alignment with the light–dark cycle, influencing synaptic activity in a phase-dependent manner [86]. This inflammatory environment disrupts REM sleep regulation by impairing inhibitory projections from the brainstem to the spinal cord, leading to increased motor activity during REM sleep [87,88]. Furthermore, chronic neuroinflammation weakens the integrity of the blood–brain barrier (BBB), allowing peripheral immune cells to infiltrate and sustain neurodegeneration.

HD and amyotrophic ALS also exhibit profound sleep disturbances linked to glial dysfunction and neurodegeneration. In HD, which is caused by mutant huntingtin (mHTT) aggregation and striatal atrophy, astrocytic glutamate transporter (EAAT2) dysfunction results in excitotoxic neuronal loss and disrupted sleep–wake cycles. HD patients exhibit fragmented sleep, reduced SWS, and altered circadian rhythms, likely due to the combined effects of astrocytic dysfunction and thalamostriatal degeneration [89]. Furthermore, microglial hyperactivation in the striatum contributes to synaptic loss and worsens motor and cognitive symptoms. In ALS, where motor neuron degeneration leads to progressive paralysis, sleep disturbances are common due to brainstem and cortical dysfunction. Astrocytes in ALS exhibit toxic gain-of-function phenotypes, releasing glutamate and pro-apoptotic factors that accelerate motor neuron death [90]. Glial-mediated excitotoxicity impairs sleep-related neural circuits in the brainstem, leading to poor sleep quality and nocturnal respiratory failure. Additionally, microglial activation in ALS drives neuroinflammation, which contributes to sleep fragmentation and rapid disease progression [91,92,93,94].

The dysregulation of astrocytes and microglia due to chronic stress and sleep disturbances plays a crucial role in the pathophysiology of psychiatric disorders, including depression, anxiety, bipolar disorder, and PTSD. Recent findings suggest that microglia likewise regulate sleep and wakefulness through CSF1R receptors, both under normal conditions and during stress. Depleting phagocytes with PLX5622 reduces wakefulness, alters sleep architecture, and amplifies the effects of psychosocial stress, underscoring their essential role in stress resilience [95,96,97]. Sleep disturbances are hallmark symptoms of these disorders, with alterations in REM sleep, NREM sleep, and sleep continuity contributing to emotional dysregulation and cognitive impairment. Chronic stress disrupts glial homeostasis, neuroinflammatory pathways, and neurotransmitter balance, exacerbating sleep dysfunction and psychiatric symptoms [98].

Depression and anxiety disorders are strongly associated with chronic stress, sleep disturbances, and dysregulated glial function. Depressed individuals often exhibit shortened REM latency, increased REM density, and disrupted NREM slow-wave activity (SWA), while anxiety disorders are frequently accompanied by hyperarousal, sleep fragmentation, and prolonged sleep onset latency [94].

Astrocytes play a pivotal role in regulating sleep homeostasis, glutamate clearance, and adenosine signaling, all of which are disrupted in depression and anxiety. As stated before, chronic stress impairs astrocytic glutamate uptake by downregulating EAAT2, leading to excessive glutamatergic activity in limbic regions such as the amygdala and prefrontal cortex. This excitotoxic environment disrupts SWS, weakens synaptic homeostasis, and promotes emotional dysregulation, key features of depression and anxiety. In addition, microglial dysfunction further contributes to sleep disturbances in these disorders. Under chronic stress, microglia shift toward a pro-inflammatory M1 phenotype, releasing TNF-α, IL-1β, and IL-6, which impair sleep stability and REM sleep regulation. Elevated IL-1β disrupts SWA and enhances REM pressure, while TNF-α inhibits orexinergic neurons, leading to increased daytime sleepiness and altered sleep–wake cycles [37,99,100,101]. These inflammatory changes contribute to depressive symptoms, cognitive dysfunction, and persistent negative effects.

Bipolar disorder (BD) is characterized by alternating manic and depressive episodes, each associated with distinct sleep disturbances and glial dysfunction. Mania is typically linked to reduced total sleep time, decreased REM latency, and increased nocturnal arousals, while bipolar depression exhibits prolonged sleep onset, hypersomnia, and increased REM density. Astrocytic dysfunction contributes to mood instability and sleep disturbances in BD by disrupting glutamate homeostasis and energy metabolism. In manic episodes, excessive glutamate release and impaired astrocytic uptake lead to hyperexcitability, reduced SWS, and fragmented REM sleep [102,103]. Conversely, in depressive episodes, astrocytic dysfunction results in synaptic atrophy, diminished neuronal plasticity, and excessive REM pressure, exacerbating emotional instability and cognitive deficits. Microglial dysregulation further worsens bipolar-associated sleep disturbances. Manic episodes are associated with microglial-driven oxidative stress and neuroinflammation, which disrupt circadian rhythms and sleep stability. Elevated TNF-α and IL-6 levels suppress NREM sleep and increase nocturnal wakefulness, mirroring the hyperarousal state seen in mania. In contrast, bipolar depression is marked by microglial-induced synaptic pruning and reduced neurotrophic support, contributing to sleep fragmentation and prolonged REM episodes [103,104,105,106].

PTSD is a stress-related psychiatric condition characterized by hyperarousal, nightmares, and fragmented sleep patterns. Patients with PTSD often exhibit reduced SWS, increased REM density, and heightened nocturnal awakenings, contributing to fear memory consolidation and emotional dysregulation [107,108]. Astrocytes in PTSD show impaired glutamate clearance and dysregulated norepinephrine (NE) metabolism, leading to persistent hypervigilance and sleep disturbances [109,110,111,112]. Chronic stress reduces EAAT2 expression in the prefrontal cortex and hippocampus, increasing excitatory neurotransmission and promoting hyperarousal [88]. Moreover, astrocytic dysfunction disrupts the glymphatic system, impairing trauma-related memory processing during sleep and worsening PTSD symptoms [31,49].

Microglial hyperactivation plays a key role in PTSD-associated neuroinflammation and synaptic remodeling. Elevated levels of pro-inflammatory cytokines (IL-1β and TNF-α) in the amygdala and hippocampus enhance fear-related neural circuits, reinforcing intrusive thoughts and nightmares [90,91]. Additionally, microglia-mediated synaptic pruning of inhibitory circuits in PTSD leads to GABAergic deficits, reducing sleep stability and increasing sleep fragmentation [92].

## 7. Key Topics and Future Directions

### 7.1. The Role of the Gut–Brain Axis: Microbiota’s Influence on Glial Function, Stress, and Sleep

The gut–brain axis (GBA) has emerged as a crucial modulator of glial activity, sleep physiology, and stress resilience, with the gut microbiota influencing astrocytic and microglial function through metabolites, immune signaling, and neural pathways. These interactions impact sleep architecture and neuroinflammation, particularly through short-chain fatty acids (SCFAs) produced by gut bacteria. SCFAs, such as butyrate, play a significant role in regulating microglial activation and astrocytic metabolism, which in turn affects sleep–wake regulation. For instance, butyrate enhances astrocytic neurotrophic support, promoting slow-wave sleep and cognitive function [113,114,115,116]. Conversely, gut dysbiosis (an imbalance in the gut microbiota, often observed in stress-related disorders) leads to increased peripheral inflammation and microglial hyperactivation, contributing to sleep disturbances and mood disorders. Moreover, the gut microbiota modulates the synthesis of key neurotransmitters like serotonin and melatonin, both essential for circadian rhythm regulation and sleep homeostasis. The microbiota affects serotonin synthesis through tryptophan metabolism, where specific bacteria like Lactobacilli and Bifidobacteria help convert tryptophan into serotonin, impacting not only gut serotonin levels but also brain serotonin regulation. SCFAs also influence melatonin production by modulating tryptophan conversion into serotonin and subsequently melatonin. Dysbiosis, particularly serotonin depletion in the gut, has been linked to insomnia, increased REM fragmentation, and heightened stress susceptibility [116]. Given these findings, probiotics, prebiotics, and dietary interventions targeting gut microbial composition are being explored as potential therapeutic strategies for stress-induced sleep disorders. Future research should focus on identifying specific microbial strains that promote glial homeostasis and sleep stability with the goal of developing novel, microbiota-based therapies for improving stress resilience and sleep health.

### 7.2. Sex Differences: Impact of Hormonal Variations on Glial Responses to Stress and Sleep Dysregulation

Sex differences play a critical role in stress susceptibility, sleep patterns, and glial function, yet they remain understudied in sleep research. Hormonal fluctuations, particularly in estrogen and progesterone levels, influence astrocytic and microglial activity, leading to sex-specific sleep disturbances in neuropsychiatric disorders [117,118]. Estrogen has been shown to modulate astrocytic glutamate uptake and neurotrophic factor expression, affecting REM sleep regulation and cognitive performance [119]. During periods of low estrogen, such as menopause or the luteal phase, astrocytic glutamate dysregulation and impaired adenosine signaling contribute to insomnia and fragmented sleep [119,120]. Microglia also exhibit sex-specific responses to stress. Female microglia tend to adopt a pro-inflammatory phenotype under chronic stress, increasing susceptibility to depression-related sleep disturbances. In contrast, male microglia show greater resilience to neuroinflammatory insults but are more prone to neurodegenerative pathologies involving sleep disruption [100]. Estrogen promotes the expression of pro-inflammatory cytokines such as TNF-α, IL-1β, and IL-6, driving microglia into a more reactive, pro-inflammatory phenotype. This heightened inflammatory response increases susceptibility to stress-induced disorders like depression and anxiety, contributing to sleep disturbances such as insomnia and REM fragmentation. Chronic stress exacerbates these effects, with estrogen amplifying microglial reactivity, which impacts brain regions involved in sleep regulation and emotional processing [121]. On the other hand, male microglia exhibit a different response, with testosterone promoting more anti-inflammatory and neuroprotective functions, making them less susceptible to chronic stress-induced neuroinflammation but more prone to neurodegenerative diseases associated with sleep disruptions [100]. Understanding these sex differences is crucial for designing personalized sleep interventions. Future research should prioritize sex-stratified clinical trials to develop targeted hormonal, glial-modulating, and anti-inflammatory treatments tailored to individual sleep disorders.

### 7.3. Future Directions

The development of high-resolution imaging techniques and single-cell transcriptomics has revolutionized the study of glial dynamics in sleep and stress responses. Emerging technologies, such as two-photon microscopy, functional MRI (fMRI), and single-cell RNA sequencing, allow researchers to map glial activity across sleep–wake cycles and identify stress-induced molecular changes at an unprecedented resolution. Two-photon calcium imaging has revealed that astrocytes exhibit sleep-dependent calcium waves, suggesting an active role in sleep regulation and synaptic maintenance [122]. Additionally, optogenetic and chemogenetic tools have enabled researchers to selectively manipulate astrocyte and microglial activity to assess their causal roles in sleep–wake regulation [123]. Recent single-cell transcriptomic studies have identified stress-induced transcriptional shifts in glial cells, including altered cytokine signaling, metabolic dysfunction, and gliotransmitter release patterns in sleep disorders [124,125]. Future research leveraging multi-omics approaches will help elucidate how glial heterogeneity and plasticity contribute to sleep disturbances in psychiatric and neurodegenerative conditions.

Recent advances in neuroscience and glial biology have revealed novel insights into how astrocytes and microglia contribute to stress-induced sleep dysregulation and its implications for neurodegenerative and psychiatric disorders. Targeting HPA axis dysregulation and its downstream effects presents a promising avenue for restoring sleep homeostasis and mitigating the negative consequences of chronic stress [33,126]. The stress-induced dysregulation of astrocytes and microglia establishes a self-reinforcing loop of excitotoxicity, oxidative stress, neuroinflammation, and synaptic dysfunction, which significantly disrupts sleep homeostasis and contributes to neurodegenerative and psychiatric disorders. Targeting these maladaptive glial responses offers a promising avenue for therapeutic intervention. Tools such as glucocorticoid receptor antagonists, CRH receptor blockers, and circadian rhythm modulators may help normalize sleep patterns and improve resilience against stress-related disorders [127].

The role of astrocytes in sleep regulation underscores their potential as therapeutic targets for sleep disorders and neurodegenerative diseases. Astrocyte-directed treatments include modulating adenosine signaling as astrocytes regulate extracellular adenosine levels to promote sleep [128]. A1 receptor agonists and adenosine reuptake inhibitors have shown potential in enhancing SWS and sleep-dependent memory consolidation. Additionally, drugs that upregulate EAAT2 can enhance glutamate clearance, preventing excitotoxicity and synaptic dysfunction associated with stress-related sleep disorders [129,130]. Pharmacological approaches aiming to restore astrocytic function, such as AQP4 modulators, adenosine receptor agonists, and glutamate reuptake enhancers, hold promise for mitigating the impact of chronic stress on sleep homeostasis. Restoring astrocytic glutamate uptake through EAAT2 enhancers (e.g., riluzole and ceftriaxone) could reduce excitotoxicity and stabilize sleep patterns [131,132]. Reducing microglial hyperactivation using anti-inflammatory agents (e.g., minocycline and IL-1 receptor antagonists) may help restore sleep architecture and mitigate cognitive decline [133]. Enhancing astrocytic antioxidant defenses through N-acetylcysteine (NAC) supplementation could protect against oxidative stress-induced neurodegeneration [134]. Modulating complement signaling to prevent excessive synaptic pruning may help preserve cognitive function and emotional resilience under chronic stress. Additionally, behavioral and lifestyle interventions that enhance glymphatic function, such as optimizing sleep posture and increasing deep sleep duration, may offer complementary strategies for maintaining cognitive health and preventing neurodegenerative progression. Understanding the astrocytic mechanisms that govern sleep homeostasis is therefore critical for developing targeted therapies aimed at preserving neural function in stress-related and age-associated disorders. Microglial-targeting therapies aim to reduce neuroinflammatory responses that disrupt sleep architecture. Inhibitors of TNF-α, IL-1β, and IL-6 have been shown to restore normal REM/NREM cycles in stress-induced sleep disturbances. Moreover, repurposing microglial modulators such as minocycline and ibudilast has demonstrated promise in reducing sleep fragmentation in depression and PTSD by attenuating chronic inflammation and excessive synaptic pruning. Understanding the precise molecular mechanisms linking chronic stress to sleep disturbances via neuroinflammation is essential for developing targeted therapeutic interventions. Strategies aimed at modulating glial activation, reducing pro-inflammatory cytokine levels, and restoring sleep architecture hold promise for mitigating the detrimental effects of stress on both sleep and neurodegenerative processes. Pharmacological approaches such as IL-1 receptor antagonists, TNF-α inhibitors, and astrocyte-specific anti-inflammatory agents, as well as behavioral interventions targeting the HPA axis, may offer novel avenues for restoring sleep integrity and improving long-term neurological outcomes [133,134,135].

Emerging evidence suggests that targeting glial-mediated neuroinflammation in PD could improve both dopaminergic neuron survival and sleep quality. Minocycline, an anti-inflammatory microglial modulator, has shown promise in restoring REM sleep architecture and reducing neuroinflammatory burden in preclinical PD models. In addition, treatments targeting glial dysfunction in PTSD could include anti-inflammatory therapies (e.g., ibudilast), astrocyte-targeting glutamate modulators, and interventions aimed at restoring glymphatic clearance [136,137]. By normalizing glial function, these strategies may help restore healthy sleep patterns and alleviate psychiatric symptoms. Advancements in gene therapy and pharmacogenomics will further enable the precise targeting of glial dysfunction in sleep disorders, opening new avenues for personalized medicine.

## 8. Conclusions

The reviewed evidence highlights the crucial roles of astrocytes and microglia in stress-induced sleep dysregulation, particularly their involvement in sleep–wake homeostasis, neuroinflammation, synaptic remodeling, and glymphatic clearance. Chronic stress disrupts the delicate balance of glial activity, leading to prolonged neuroimmune activation, oxidative stress, and neurotransmitter dysregulation, which, in turn, contribute to fragmented sleep, REM/NREM imbalances, and heightened vulnerability to neurodegenerative and psychiatric disorders.

Astrocytes play a fundamental role in sleep homeostasis, regulating adenosine signaling, glutamate uptake, and metabolic waste clearance. Their dysfunction under chronic stress conditions impairs synaptic plasticity, glymphatic flow, and neuromodulatory balance, exacerbating cognitive decline and emotional instability. Similarly, stress-induced microglial hyperactivation drives chronic neuroinflammation and excessive synaptic pruning, leading to disruptions in sleep architecture, increased neuronal vulnerability, and higher susceptibility to disorders such as AD, PD, depression, PTSD, and bipolar disorder.

Despite significant advances in understanding the glial–sleep–stress axis, translational research in this area remains in its early stages. Targeting glial dysfunction presents a promising avenue for novel therapeutics aimed at restoring sleep integrity and mitigating stress-induced neuropathology. Future research should focus on developing pharmacological and non-pharmacological interventions that modulate astrocytic and microglial activity, including astrocyte-specific adenosine modulators, microglial anti-inflammatory agents, and gut microbiome-based therapies.

Additionally, integrating cutting-edge imaging techniques, multi-omics approaches, and computational modeling will be essential for deciphering the precise molecular pathways underlying glial-mediated sleep dysregulation. Understanding sex-specific differences in glial function, the influence of circadian disruptions, and the long-term impact of chronic stress on glial–sleep interactions will further refine therapeutic strategies.

Given the widespread impact of stress-induced sleep disturbances on cognitive health, emotional stability, and disease progression, prioritizing glia-focused translational research is critical. Bridging the gap between basic neuroscience and clinical applications will enable the development of targeted interventions to enhance sleep quality, reduce neuroinflammatory damage, and improve overall brain health in individuals suffering from stress-related sleep disorders.

## Figures and Tables

**Figure 1 biomedicines-13-01121-f001:**
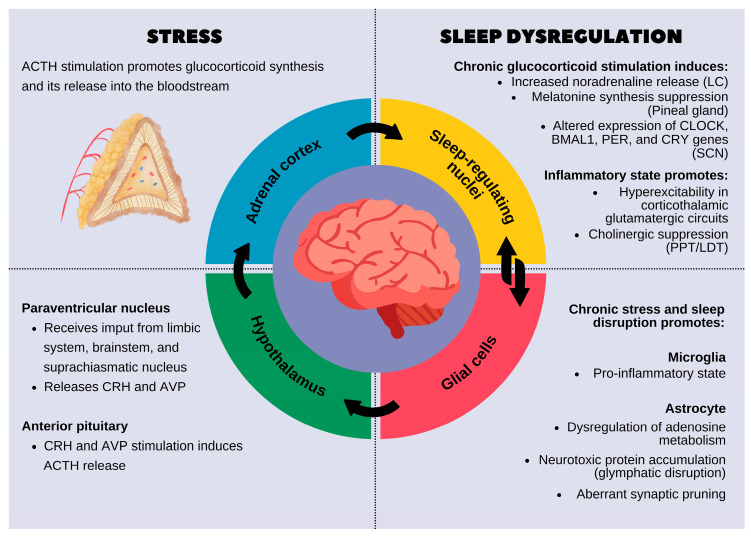
**The reciprocal relationship between stress and sleep dysregulation.** On the left, the stress response is shown, beginning with the activation of the hypothalamic–pituitary–adrenal (HPA) axis, leading to glucocorticoid release from the adrenal cortex. Chronic activation of this system affects sleep-related brain regions and disrupts normal sleep processes. On the right, chronic stress-induced sleep dysregulation is detailed. Prolonged glucocorticoid exposure alters the function of sleep-regulating nuclei by increasing noradrenaline levels, suppressing melatonin production, and modifying circadian gene expression in the suprachiasmatic nucleus (SCN). Inflammatory processes further disrupt sleep by promoting hyperexcitability in the corticothalamic glutamatergic circuits and suppressing cholinergic signaling. Additionally, glial cells, including microglia and astrocytes, play a key role in mediating the effects of stress on sleep. Chronic stress promotes a pro-inflammatory microglial state and impairs astrocytic functions, leading to the dysregulation of adenosine metabolism, glymphatic clearance failure, and abnormal synaptic pruning. These changes contribute to fragmented, poor-quality sleep, which in turn exacerbates stress, creating a vicious cycle.

**Figure 2 biomedicines-13-01121-f002:**
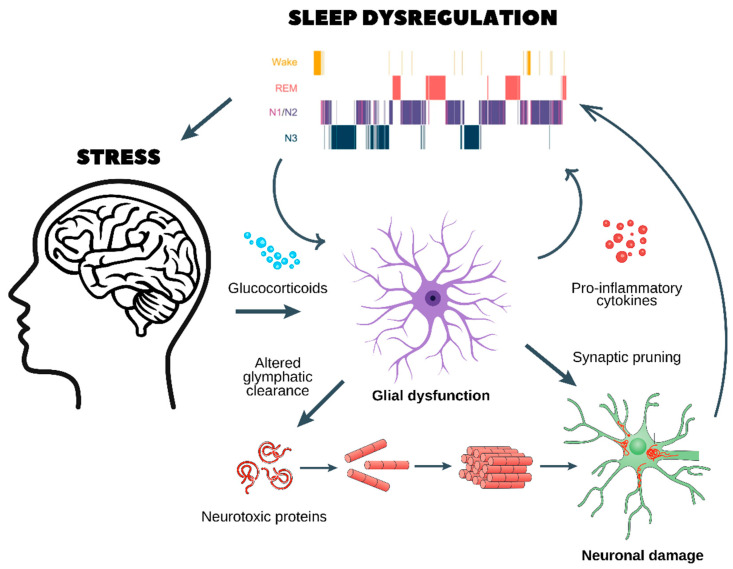
Chronic stress, glial dysfunction, and sleep dysregulation: a pathway to neurodegeneration and psychiatric disorders. This figure illustrates the pathological cycle through which chronic stress disrupts sleep homeostasis, leading to glial dysfunction, neuroinflammation, and neuronal damage, which in turn contribute to the development of neurodegenerative and psychiatric disorders.

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
