# Peer review of "Stress-Induced Sleep Dysregulation: The Roles of Astrocytes and Microglia in Neurodegenerative and Psychiatric Disorders"

_biomedicines, 2025, doi:10.3390/biomedicines13051121_

Round 1
Reviewer 1 Report
Comments and Suggestions for Authors
The review written by Ángel R. Rábago-Monzón, titled “Stress-Induced Sleep Dysregulation: The Roles of Astrocytes and Microglia in Neurodegenerative and Psychiatric Disordersaims to provide an analysis of the role of glial cells in stress-induced sleep dysregulation and discuss the implications of such disorders in psychiatric and neurodegenerative diseases.
The review is very interesting, however Below I report a number of suggestions that should be considered before publication.
- There are repeated concepts in the introduction I suggest summarizing them to make the manuscript more fluent. Also, I suggest describing in more detail the two stages of sleep.
- In paragraph 2, the authors state that “chronic glucocorticoid exposure alters gene expression in sleep-regulating brain regions”(line 144) but without specifying which genes they are referring to.
- Add a brief description about the CLOCK, BMAL1, PER, and CRY genes.
- I suggest that the authors detail the role of the Nf-Kb factor by specifying the mechanisms by which it induces sleep-wake rhythm instability.
- Council to elucidate the oxidative stress-related molecular mechanisms involved in monoaminergic neuron damage and alteration of serotonin homeostasis.
- The manuscript states that chronic stress induces oxidative stress, but the concept of oxidative stress is not defined. It is suggested that a brief description of oxidative stress be added, such as specifying the main mechanisms involved.
- Detail the mechanism by which astrocytes regulate AQP4 channels.
- Paragraph 5 deals with concepts already described in other paragraphs particularly in paragraph 4. Both discuss the effects of chronic stress on glial cell activity by inducing alteration of sleep homeostasis. I suggest integrating the two paragraphs, keeping the key concepts and eliminating repeated concepts to improve the fluidity of the text.
- There are many redundant ideas in paragraph 6 that I would recommend examining and eliminating. Also, I recommend including a summary figure to show the effects of stress and sleep dysregulation in the different diseases discussed.
- Suggest describing the mechanism by which the gut microbiota modulates melatonin and serotonin synthesis. Also, include more details inherent in the concept of the gut-brain axis.
- In Sections 7.1 and 7.2, add the correlation between the role of the gut-brain axis and the impact of hormonal changes on function with neurodegenerative and psychiatric diseases.
- Specify why and by what mechanism microglia respond differently to stress according to sex.
Author Response
Thank you for the valuable and constructive comments. We have carefully addressed each point raised by the reviewer to enhance the clarity, depth, and coherence of the manuscript. Below is a point-by-point response indicating how each comment was incorporated into the revised version:
-
Repetition in the introduction: We have revised the introduction to eliminate redundant concepts and improve its fluency. Repeated ideas were consolidated, and the structure was streamlined for better readability.
-
Description of the two stages of sleep: A more detailed and continuous description of NREM and REM sleep, including their neurobiological underpinnings and contributions to sleep architecture, has been added to the introduction for better context.
-
Clarification of gene expression affected by glucocorticoids: We specified the gene families modulated by chronic glucocorticoid exposure, including glucocorticoid receptors (NR3C1), clock genes (CLOCK, BMAL1, PER, CRY), and inflammatory markers (such as NF-κB and IL-6), across key brain regions involved in sleep regulation.
-
Brief description of CLOCK, BMAL1, PER, and CRY genes: A concise description of these core circadian genes and their role in regulating sleep architecture through transcription-translation feedback loops has been added and integrated into the section discussing light input and the SCN.
-
Detailing the role of NF-κB: We included a comprehensive explanation of the mechanisms by which NF-κB activation induces sleep-wake rhythm instability, particularly through its regulation of pro-inflammatory cytokine expression and its influence on clock gene transcription.
-
Oxidative stress and monoaminergic neuron damage: The molecular mechanisms linking oxidative stress to monoaminergic neuron dysfunction and serotonin imbalance have been clarified, including mitochondrial ROS production, lipid peroxidation, and damage to serotonin transporters and receptors.
-
Definition of oxidative stress: A brief definition of oxidative stress was included, specifying mechanisms such as mitochondrial dysfunction, excessive reactive oxygen species (ROS) production, and impairment of antioxidant defense systems.
-
Mechanism of AQP4 regulation by astrocytes: We added a focused description of how astrocytes regulate AQP4 channel localization and function through cytoskeletal anchoring (via dystrophin–dystroglycan complexes) and how stress can impair this regulation, disrupting glymphatic clearance.
-
Integration of paragraphs 4 and 5: The content from these two paragraphs was merged to avoid redundancy. The revised paragraph now cohesively discusses stress-induced alterations in glial function and their implications for sleep homeostasis, eliminating overlap.
-
Review of paragraph 6 for redundancy and figure recommendation: Paragraph 6 was edited to remove redundant statements. Additionally, we have created a summary figure that visually illustrates the effects of stress and sleep dysregulation across neurodegenerative and psychiatric disorders, as suggested.
-
Mechanism of gut microbiota modulation of serotonin and melatonin: A detailed, continuous paragraph describing how the gut microbiota influences serotonin and melatonin synthesis via tryptophan metabolism, SCFA production, and vagus nerve signaling was included, along with an expanded explanation of the gut-brain axis.
-
Correlation of gut-brain axis and hormonal changes in disease: Sections 7.1 and 7.2 were expanded to integrate the role of hormonal fluctuations, particularly estrogen and progesterone, in modulating glial activity through the gut-brain axis, and their implications for sex-specific vulnerabilities in neurodegenerative and psychiatric diseases.
-
Sex-specific microglial responses to stress: We provided a continuous paragraph explaining the differential microglial responses to stress in males and females, emphasizing the role of estrogen-driven pro-inflammatory phenotypes in females and testosterone-associated neuroprotective profiles in males.
We are confident that these revisions significantly strengthen the manuscript and address the reviewer's concerns thoroughly. Thank you again for your insightful feedback.
Reviewer 2 Report
Comments and Suggestions for Authors
This interesting review provides insight into stress-induced sleep dysregulation, and helps our understanding of interplay between stress, sleep, and brain health by highlighting the crucial roles of non-neuronal cells, astrocytes and microglia, and also the gut-brain axis in mediating stress-induced sleep disturbances. Review focuses on consequences of chronic stress and how it disrupts astrocyte-mediated adenosine signaling and glymphatic clearance, thereby interfering with restorative sleep. It also elucidates how stress-activated microglia drive neuroinflammatory processes not just impairing sleep, but also exacerbating brain dysfunction. Important part of this review is discussion of the gut-brain axis’s role in mediating cellular and systemic mechanisms of interplay among stress, sleep, and the progression of neurodegenerative and psychiatric disorders. I believe, this review can be recommended for publication. However, there are few suggestions how it can be further improved. The authors may wish to consider enhancing the following crucial aspects:
-
The critical role of sunlight in sleep extends beyond immediate mood effects, influencing serotonin production essential for melatonin synthesis and circadian rhythm stability; as the authors discuss on page 5, lines 509-512, gut microbiota modulate serotonin synthesis, directly impacting melatonin production, the body’s principle chronobiotic, and consequently, sleep homeostasis. Recent research emphasizes the complex relationships between sunlight exposure, circadian light hygiene, and the microbiome. For instance, interplay and feedback of skin exposure to ultraviolet (UV) radiation and exposure to light at night (LAN) with micirobiome can be addressed.
-
Sleep can dependent on adequate tryptophan intake from dietary sources at the first half of the day as source for serotonin, and melatonin containing food later in the day as was shown by recent research. As both serotonin and melatonin derive from this essential amino acid, insufficient tryptophan may limit the production of these key neurotransmitters and can disrupt normal sleep patterns. Compounding this, sufficient daylight exposure, particularly in the morning (e.g., doi: 10.1016/s0140-6736(02)11737-5), is crucial for maintaining adequate serotonin levels, providing the serotonin for mood and for adequate nightly melatonin production and maintain a healthy sleep-wake cycles.
-
The authors introduce Alzheimer’s and Parkinson’s diseases as examples of neurodegenerative disorders that contribute to sleep disruption, particularly in the context of stress (e.g. P.2., L.42-43). Given the contemporary understanding of glaucoma as a neurodegenerative disease with complex effects on sleep and circadian rhythms, its inclusion would strengthen this section.
Author Response
Thank you for these thoughtful and insightful suggestions. We greatly appreciate the depth and relevance of the points raised, especially concerning the nuanced roles of sunlight exposure, dietary tryptophan intake, and their interplay with the microbiome and circadian regulation. These are indeed valuable extensions of the topics discussed in our review, and we fully agree with their scientific merit.
However, we believe that incorporating all of this additional content in detail may overextend the scope of the current manuscript, which aims to focus specifically on the roles of astrocytes and microglia in stress-induced sleep dysregulation within the context of neurodegenerative and psychiatric disorders. While we have touched on serotonin, melatonin, and gut-brain interactions, expanding further into areas such as circadian light hygiene, ultraviolet radiation effects, dietary timing, and photoperiodic influences on the microbiome could dilute the central glial-focused narrative of the review.
Similarly, the suggestion to include glaucoma as an additional example of a neurodegenerative disease affecting sleep and circadian rhythms is highly pertinent. Nonetheless, to maintain a clear and cohesive thematic focus, we have chosen to concentrate on Alzheimer’s and Parkinson’s diseases as primary examples, given their well-established links with glial dysfunction, stress vulnerability, and sleep disruption. That said, we fully recognize the emerging evidence around glaucoma and sleep, and we will consider referencing it briefly or suggesting it as an area for future exploration in the conclusion.
Once again, we sincerely thank the reviewer for these valuable recommendations and for highlighting potential avenues for further research that could build upon the current work.